# Fatigue Crack Propagation of 51CrV4 Steels for Leaf Spring Suspensions of Railway Freight Wagons

**DOI:** 10.3390/ma17081831

**Published:** 2024-04-16

**Authors:** Vítor M. G. Gomes, Grzegorz Lesiuk, José A. F. O. Correia, Abílio M. P. de Jesus

**Affiliations:** 1FEUP, Faculty of Engineering, University of Porto, R. Dr. Roberto Frias, 4200-465 Porto, Portugal; jacorreia@fe.up.pt (J.A.F.O.C.); ajesus@fe.up.pt (A.M.P.d.J.); 2Wroclaw University of Science and Technology, Wybrzeże Stanisława Wyspiańskiego 27, 50-370 Wrocław, Poland; grzegorz.lesiuk@pwr.edu.pl

**Keywords:** railway, rolling stock, freight wagon, leaf springs, fatigue crack growth, fracture surfaces

## Abstract

Leaf springs are critical components for the railway vehicle safety in which they are installed. Although these components are produced in high-strength alloyed steel and designed to operate under cyclic loading conditions in the high-cyclic fatigue region, their failure is still possible, which can lead to economic and human catastrophes. The aim of this document was to precisely characterise the mechanical crack growth behaviour of the chromium–vanadium alloyed steel representative of leaf springs under cyclic conditions, that is, the crack propagation in mode I. The common fatigue crack growth prediction models (Paris and Walker) considering the effect of stress ratio and parameters such as propagation threshold, critical stress intensity factor and crack closure ratio were also determined using statistical methods, which resulted in good approximations with respect to the experimental results. Lastly, the fracture surfaces under the different test conditions were analysed using SEM, with no significant differences to declare. As a result of this research work, it is expected that the developed properties and fatigue crack growth prediction models can assist design and maintenance engineers in understanding fatigue behaviour in the initiation and propagation phase of cracks in leaf springs for railway freight wagons.

## 1. Introduction

The industry provides a wide range of spring steels with variable properties for a variety of technological applications. The chromium–vanadium alloyed steel, 51CrV4 steel grade, has been used for the design of industry machinery components and suspension elements of road and rail vehicles [1,2,3]. In the rail freight sector, 51CrV4 steel grade is usually found in leaf springs due to its high mechanical strength to static and fatigue loadings [4,5].

Leaf springs, as critical components for vehicle safety, are often designed for an infinite life; however, occasionally, a fatigue fracture occurs in these components (see Figure 1). Since this component is designed to operate in the high-cyclic fatigue regime, the fatigue phenomenon is expected to be governed mostly by surface crack initiation mechanisms. Nevertheless, to maximize the availability of the vehicle fleet, inspection and maintenance periods of the components need to be optimized. Then, the optimization of the intervention strategy can be performed by knowing the percentage of life associated with the crack propagation life. Furthermore, fatigue failure of suspension elements can lead to rolling stock derailment, leading to economic losses [6].

Different types of investigations have been carried out in order to understand the resistance to the fatigue crack growth in leaf springs. These studies have been carried out at both the material level and the component level.

The investigation regarding fracture resistance properties in leaf springs is of great importance, not only due to their quasi-brittle behaviour [4], but also because thermal and mechanical treatments can significantly change these properties. For example, the number of cycles for fatigue failure tends to be higher when the tempering temperature increases [4,7]. These variations may be greater if a superficial shot-peening treatment is applied. When this mechanical treatment is applied, the fatigue crack can start from internal inclusions (oxides, carbides) or surface defects (due to its high surface roughness [8,9]), which then changes both the number of cycles for failure and threshold propagation limit [10,11,12]. Concerning fracture toughness, this material property tends to progressively decrease with the decrease in tempering temperature, but it has an inverse evolution in relation to the hardness of the material [11,13,14].

Regarding the quenching treatment, increasing the quenching temperature from 840 to 920 °C, it was observed in [11] an increase in the speed rate of crack propagation, in the regime II of propagation, for the propagation mode I. In addition, the quenching and tempering process affect the crack propagation behaviour. According to investigations carried out on low-alloy bainitic steel (51CrV4), the effect of heat treatment is beneficial for the propagation threshold value [15]. Testing a 51CrV4 low-alloy bainitic steel for different stress ratios and propagation directions, a significant influence was found for both cases in propagation regime I; however, this effect started being irrelevant as the crack entered in propagation regime II.

Via scanning electron microscopy (SEM), transgranular fracture surfaces were observed. In these fractures’ surfaces, secondary cracks and fracture micro-mechanism, especially due to facet cleavage, were observed [7,11,16]. Additionally, reduced-sized ductile dimples were observed. Regarding the development of fatigue striations, their visualization is not frequently observed; however, when they were observed in martensitic steels, on a very small scale, according to [7], the fatigue striations had quite a reduced spacing (around 1 μm).

Concerning the type of fracture surface frequently observed in 51CrV4 steel, the fatigue crack propagation process is transgranular, because in cases where intergranular fracture surfaces were observed in leaf springs, these cracks originated essentially from the occurrence of corrosion pitting that promoted environmental embrittlement, and hence cracks propagated between grain boundaries [17].

Thus, combining the material complexity (associated with the production of leaf springs) with the importance of structural safety that these components have, in this research, a crack growth analysis in cyclic conditions of spring steel for parabolic springs applied to rail freight was carried out. Regardless of the leaf spring geometry, fracture surfaces observed in leaf springs tend to exhibit a perpendicular propagation direction to the maximum principal stress, showing that crack propagation is perfectly defined by the mode I propagation direction (crack propagation perpendicular to the loading) [10,16,17,18,19,20,21,22]. Thus, the fatigue crack growth properties of the material were determined only for mode I, using compact tension (CT) specimens. Initially, the Paris law for propagation regime II and the threshold law for propagation regime I were determined via the least-squares method for distinct stress ratios. Additionally, the crack closure effect was analysed via the crack closure ratio. The fatigue crack growth analysis was extended using Walker’s model to represent a full-range relationship. In this full-range model, Walker’s parameter was used to combine the propagation behaviour in regimes I and II under distinct stress ratios. At last, a fracture surface analysis using scanning electron microscopy (SEM) was performed. The effect of the stress ratio was evaluated on the topography of the sampled surfaces.

The crack propagation properties and crack propagation curves determined and developed in this research are essential for understanding the crack propagation behaviour in chromium–vanadium alloyed steel, 51CrV4. Furthermore, it is expected that the properties obtained from the analyses carried out in this document can be used in further damage-tolerant approaches along with numerical simulations to predict the fatigue life of leaf springs, with the aim of avoiding serious losses associated with the failure of suspension elements, namely, leaf springs.

## 2. Fatigue Crack Growth

After the fatigue crack initiation stage is reached, a fatigue crack with a macro-size propagates steadily until reaching its critical size, leading to its structural failure. This steady crack growth has been described in terms of controlled-failure assumptions from fracture mechanics theory. In the case of materials exhibiting a linear elastic behaviour or a low amount of plasticity around the crack tip, LEFM (linear elastic fracture mechanics) is applicable. LEFM suggests that the stress state at the crack tip be represented by the stress intensity factor, *K*.

### 2.1. Fatigue Crack Propagation Behaviour

Under cyclic loadings, LEFM is still suitable for fatigue crack propagation analysis if the stress intensity factor range, ΔK, is assumed. The subcritical crack propagation behaviour (regime II) of spring steel is well represented [12,22,23,24] by Paris’s law (Equation (Equation 1)) [25,26], such that
(1)$$\frac{da}{dN}=C{\left(\Delta K\right)}^{m},$$
where *C* and *m* are material constants determined by experimental data. The stress intensity factor range, $\Delta K=K_{max}-K_{min}$, defined for a cycle and related to the stress range, Δσ, and crack length, *a*, is given by Equation (Equation 2)
(2)$$\Delta K=Y\Delta \sigma \sqrt{\pi a},$$
with *Y* denoting a geometrical parameter.

Besides regime II, when the value of ΔK is low enough, Equation (Equation 1) may be not suitable for describing the propagation rate. Under these conditions, the crack propagation regime is identified as regime I, where a threshold value for ΔK may be observed, which means if ΔK is inferior to the threshold stress intensity factor range, $\Delta K_{th}$, no crack propagation is observed. In this regime, a power law (Equation (Equation 3)) is used to predict the da/dN−ΔK relationship for several steels, including spring steels [10], such that
(3)$$\frac{da}{dN}=A_{th}{\left(\Delta K-\Delta K_{th}\right)}^{p_{th}}.$$
with $A_{th}$ and $p_{th}$ denoting the respective regressors, also determined by experimental data.

On the other hand, at propagation regime III, when the maximum stress intensity factor value approximates the critical stress intensity factor, $K_{c}$ or the fracture toughness, $K_{Ic}$, a fast acceleration of crack propagation is observed, rapidly leading to the collapse.

### 2.2. Mean Stress Effect

As mean stress has a significant influence on fatigue life approaches, it is also expected that this variable influences the fatigue crack growth. Increasing the mean stress, the crack propagation rate, da/dN, tends to increase in all regimes, but with less impact in regime II. At regime III, as this regime is dependent on the fracture toughness of the material, substantial shifts in the crack propagation rate occur. However, regime I is the most affected region, highlighting the high influence of mean stress in the threshold stress intensity factor range, beyond the material dependence.

Several models taking into account the mean stress effect and fracture materials’ properties have been proposed based on Equation (Equation 1). Walker adapted Equation (Equation 1) for different stress ratios, $R_{\sigma }$, in regime II [27], such that
(4)$$\frac{da}{dN}=C_{w,II}{\left(\frac{\Delta K}{{\left(1-R_{\sigma }\right)}^{1-\gamma }}\right)}^{m_{w,II}}=C_{w,II}{\left(\bar{\Delta K}\right)}^{m_{w,II}}.$$
where $\bar{\Delta K}$ denotes an equivalent value of the stress intensity factor range, and the coefficient $C_{w,II}$ and the exponent $m_{w,II}$ are fitted to experimental data. γ varies between 0.3 and 1 for most metals but has typical values around 0.5 [27]. γ is dependent on the material and is directly related to the stress ratio effect, indicating a higher influence of $R_{\sigma }$ in fatigue crack growth behaviour for lower values. In the absence of experimental data, Walker’s parameter may be estimated from the ultimate tensile strength, $\sigma_{uts}$, of the material by the following equation [28]: (5)$$\gamma =-0.0002\sigma_{uts}+0.8818.$$

In addition to the Walker model, Forman combined the mean stress effect with regimes II and III [29,30], resulting in (Equation (Equation 6)), with
(6)$$\frac{da}{dN}=\frac{C_{f}{\left(\Delta K\right)}^{m_{f}}}{\left(1-R_{\sigma }\right)K_{c}-\Delta K},$$
where $C_{f}$ and $m_{f}$ are also empirical parameters fitted by available experimental data.

Along with the wide applicability of the Walker model (Equation (Equation 4)), and the importance of regime I of crack propagation, Walker combined both regimes I and II, which resulted in
(7)$$\frac{da}{dN}=C_{w,I,II}{\left(\bar{\Delta K}\right)}^{m_{w,I,II}}=C_{w,I,II}{\left(\frac{\Delta K}{{\left(1-R_{\sigma }\right)}^{1-\gamma }}-\Delta K_{th}\right)}^{m_{w,I,II}},$$
in which $\Delta K_{th}$ is described as a function of γ, such that
(8)$$\Delta K_{th}=\Delta K_{th_{\left(R_{\sigma }=0\right)}}{\left(1-R_{\sigma }\right)}^{1-\gamma },$$
results in a new Walker equation that is independent of the value of $\Delta K_{th}$ chosen, such that
(9)$$\frac{da}{dN}=C_{w,I,II}{\left(\frac{\Delta K}{{\left(1-R_{\sigma }\right)}^{1-\gamma }}-\Delta K_{th(R=0.0)}{\left(1-R_{\sigma }\right)}^{1-\gamma }\right)}^{m_{w,I,II}},$$
where the only parameters to be determined are $C_{w,I,II}$ and $m_{w,I,II}$. Notice that in Equation (Equation 9), $\Delta K_{th_{\left(R_{\sigma }=0\right)}}$ is a known value determined previously via Equation (Equation 8) using different $\Delta K_{th}$ data, which was obtained for different stress ratios.

### 2.3. Crack Closure

The importance of crack closure was identified by Elber [31], who showed that a fatigue crack may be closed even under tensile loading. Residual compressive deformation in the vicinity of the crack tip is responsible for the reduction in crack tip driving force, causing the contact of crack faces before the minimum loading is reached. The crack closure is often accounted for as a closure ratio, *U*, indicating the portion of loading in which the crack is open, as
(10)$$U=\frac{\Delta K_{eff}}{\Delta K}=\frac{1-\left(K_{op}/K_{max}\right)}{1-R_{\sigma }}.$$
with U=1 denoting that there is no closure effect, and U<<1 denoting a large closure effect. In Equation (Equation 10), $\Delta K_{eff}$ is $K_{max}-K_{op}$ and may be seen as the effective crack driving force, which is less than or equal to the nominal crack tip driving force ΔK. $K_{op}$ is the stress intensity factor in the opening, which may be equal to or slightly greater than $K_{cl}$, designated as the closure stress intensity factor.

The literature reports that the crack closure effect may have a greater effect in regime I of fatigue crack propagation. Many advances in the crack closure theory have been proposed, such as plasticity-, oxide-, or roughness-induced closure [32,33]. The plasticity-induced closure theory considers the cyclic plastic zone at the crack tip and also a wake of plasticity in deformed material along crack faces. This model is mathematically expressed by Newman’s closure Equation (Equation 11), isolating the quantity of $\Delta K_{eff}$ from Equation (Equation 10) [34]
(11)$$\Delta K_{eff}=\frac{1-f}{1-R_{\sigma }}\Delta K.$$

## 3. Material and Experimental Procedure

### 3.1. Chemical Composition and Microstructure

The steel under investigation was the chromium–vanadium alloyed steel 51CrV4 with an average carbon content of roughly 0.50% as presented in Table 1. This steel grade being standardised to be quenched at 850 °C (40 min) in an oil bath and then tempered at 450 °C for 90 min, the 51CrV4 steel (as received) exhibited a tempered martensite microstructure with retained austenite (white phases) [12] as shown in Figure 2.

### 3.2. Material and Specimen Geometry

In terms of mechanical strength under monotonic loading conditions, the statistical values were obtained from the proper tests presented in the reference [4], which followed the ISO 6892-1 [35] standards as presented in Table 2. The results refer to a batch of several specimens obtained from different spring leaves in their longitudinal and transverse directions. Table 2 shows a spring steel with high mechanical strength, $\sigma_{y}$ = 1271.48 MPa and $\sigma_{uts}$ = 1438.5 MPa, but with low (conventional) ductility at fracture, $\varepsilon_{f}$ = 7.53%. This spring steel grade exhibited a Vickers hardness of 447 HV (corresponding ≈ 45 HRC).

Regarding the analysis of the fatigue crack propagation behaviour, the propagation tests were carried out in propagation mode I, following the guidelines from the ASTM E647 standard [36]. Compact tension (CT) specimens were manufactured according to the guidelines from the ASTM E647 standard [36], resulting in the specimen geometry with a milled surface illustrated in Figure 3.

The CT geometry was manufactured by ensuring that the thickness, *B*, should be within the interval range W/20≤B≤W/4, with *W* denoting the maximum horizontal length that the crack can achieve. The initial crack, $a_{o}$, measured from the loading line, was considered to be greater than 0.2W, such that the calculation of the stress intensity factor, *K*, was not affected by small variations in the location and dimensions of the loading pin holes. The elasticity condition, $\left(W-a\right)\geq \frac{4}{\pi }\left(K_{max}/\sigma_{y}\right)$, with $K_{max}$ denoting the maximum stress intensity factor applied during the test, was also guaranteed. Table 3 presents the average and standard dimensions of the specimens used in the experimental campaign, where the nomenclature was in accordance with the ASTM E647 standard [36] and can be viewed in Figure 3.

Each specimen was obtained from different leaves belonging to distinct leaf springs. Since it is impossible to test CT specimens whose crack propagates perpendicularly to the longitudinal direction of the longitudinal axis of the leaf, as it is normally verified in the fracture surfaces of leaf springs [18,19], CT specimens were manufactured in the other directions. Thus, any effect associated with the direction of manufactured specimens in the crack propagation behaviour could be detected and then extrapolated to the crack propagation behaviour through the thickness direction. Specimens identified as LT denoted that the crack propagated through the transversal direction of the leaf. On the other hand, specimens marked as TL denoted a face of crack propagation through the longitudinal direction of the leaf’s axis. Figure 4 illustrates the directions from which the samples were taken and the respective labels.

The sample marking system was in accordance with the batch and leaf from which the sample (first and second parameters) was taken, the crack propagation directions, TL and LT, were identified by the third and fourth parameters. The fifth parameter was used to identify samples obtained by the same batch, same crack propagation direction, and tested under the same stress ratio conditions. The last three parameters (sixth to eight) were identifiers of the stress ratio used in the test (stress ratio of 0.1, 0.3, 0.5, and 0.7).

### 3.3. Apparatus and Experimental Procedure

Concerning the apparatus, the design of the grips, fixtures, and loading pins for testing CT specimens made of high-strength steels followed the ASTM E647 standard [36]. The fatigue crack propagation tests were carried out in an MTS 810 testing machine equipped with an MTS clevis gripping system to measure the crack opening displacement.

Fatigue crack propagation tests were conducted to obtain the properties of crack growth in different load ratios, $R_{\sigma }$ = 0.1, 0.3, 0.5, and 0.7, along the propagation regimes I, II, and III. Firstly, the crack propagation phase II was investigated until the unstable propagation failure and thus both behaviours in regimes II and III were gathered from a single test. The determination of the critical stress intensity factor, $K_{c}$, was made by considering the critical crack length, $a_{c}$, corresponding to a value of 95% of the crack length in the failure $a_{f}$. Posterior fatigue crack propagation tests were performed to determine the threshold stress intensity range, $\Delta K_{th}$. The procedure consisted in reducing progressively the applied value of ΔK until the stabilisation phase of ΔK by the continuous evaluation of the crack propagation rate value, da/dN.

Before the initiation of proper fatigue crack growth tests, a pre-crack of 10 mm (approximately) was made in each CT specimen in fatigue conditions with a sinusoidal waveform cyclic loading at a frequency of 12 Hz. The average value measured for the initial crack was $a_{0}=10.20\pm 0.31$ mm. The proper tests were conducted until the fracture was under conditions of constant force amplitude and a controlled increasing ΔK. The testing control was conducted by using software integrated with the MTS system and managed by a FlexTest console. The crack length measurement was performed by the compliance method according to the ASTM E647 standard [36]. Regarding the stress intensity factor range, ΔK was calculated using the dimensionless crack length, $a^{*}$, such that
(12)$$\Delta K=\frac{\Delta F}{B\sqrt{W}}\frac{\left(2+a^{*}\right)}{{\left(1-a^{*}\right)}^{3/2}}\left(0.886+4.64a^{*}-13.32a^{{*}^{2}}+14.72a^{{*}^{3}}-5.6a^{{*}^{4}}\right),$$
where $a^{*}=a/W$. The determination of da/dN was made by considering the incremental polynomial method [36], which involves the fitting of a second-order polynomial curve for a set of 2n+1 successive points, with *n* denoting the number of points. The gathered data were still used to determine the crack closure effect associated with the material. Its determination was verified by computing the crack opening force value via the compliance offset method.

### 3.4. Statistical Techniques

The fatigue crack growth models presented in this paper were usually calibrated using linear regression methods, whose parameters were estimated by the ordinary least-squares method [37] as suggested by the ASTM E647 standard [36]. Thus, the linear response function, with the vector of independent variables x and the vector of dependent variables y was written as
(13)$$\hat{y}=\beta_{0}+\beta_{1}x,$$
where the estimator $\beta_{1}$ was given by: (14)$$\beta_{1}=\frac{n\sum_{i=1}^{n_{S}}y_{i}x_{i}-\left(\sum_{i=1}^{n_{S}}y_{i}\right)\left(\sum_{i=1}^{n_{S}}x_{i}\right)}{n_{S}\sum_{i=1}^{n_{S}}x_{i}^{2}-{\left(\sum_{i=1}^{n_{S}}x_{i}\right)}^{2}}$$
and the estimator $\beta_{0}$ was explicitly determined by considering $\beta_{1}$, and the sample average values for the dependent variable, $\bar{y}$, and independent variable, $\bar{x}$, such that
(15)$$\beta_{0}=\bar{y}-{\hat{\beta }}_{1}\bar{x}.$$
Notice that in the cases where the material response function was described by a power law, the logarithm was applied to the random variables.

## 4. Results and Discussion

The results obtained are assessed and discussed throughout this section. Initially, the fatigue crack growth characterisation was conducted by considering the propagation regime II (assuming Paris’s law (Equation (Equation 1)), by analysing the effect of the rolling direction and the stress ratio, as well as the crack closure effect. Then, the properties $K_{c}$ and $\Delta K_{th}$ were determined. Finally, the calibration of the fatigue crack propagation model using the Walker model ((Equations (Equation 4) and (Equation 7)) was considered.

### 4.1. Rolling Direction Effect

Analysing the different directions of the tested specimens, LT and TL, as shown in Figure 4, it was observed that LT specimens tended to exhibit a greater value of *C*; however, LT specimens appeared to have a lower propagation speed, as seen in Table 4. The regression analysis of Equation (Equation 1) revealed for TL specimens a coefficient *C* = 3.526 $\times \;10^{-8}$± 1.827 $\times \;10^{-8}$ (mm/cycle) MPa$\sqrt{m}$ and an exponent *m* = 2.299 ± 0.1377, whereas for specimens manufactured in the LT direction, we obtained *C* = 7.534 $\times \;10^{-8}$± 3.761 $\times \;10^{-8}$ (mm/cycle) MPa$\sqrt{m}$ and *m* = 2.006 ± 0.1249. The effect of parameters *C* and *m* are visible in Figure 5.

Analysing the data presented in Table 4, one verifies that despite the existence of differences associated with the direction of the CT specimens, the differences were significant, rounding to 53% for the *C* coefficient and 15% for the exponent *m*. These differences may also be associated with some variation in mechanical properties. Thus, the following analysis was performed by considering the combined crack growth properties independent of the testing direction of the material.

### 4.2. Stress Ratio Effect and Crack Closure

The variation in crack propagation rate in regime II with the evolution of ΔK for different $R_{\sigma }$ is illustrated in Figure 6. For the stress ratios of 0.1 and 0.3, an increase in the value of the coefficient *C* and *m* with the stress ratio was observed; however, for the higher ratios, 0.5 and 0.7, the propagation rate was lower. According to the regression model (Equation (Equation 1)), the coefficients *C* and exponents *m* were very close. For $R_{\sigma }=0.1$, the material exhibited a coefficient *C* = 7.65 $\times \;10^{-8}$ and exponent m=2.03, and for $R_{\sigma }=0.5$, the material exhibited a coefficient *C* = 7.33 $\times \;10^{-8}$ and exponent m=2.06. In the case of a stress ratio of 0.5, the material exhibited a higher value of *C*, 8.78 $\times \;10^{-8}$, but the slope was slightly lower than 1.90. On the other hand, for $R_{\sigma }=0.7$, *C* = 4.39 $\times \;10^{-8}$ and m=2.15 were determined. Table 5 presents in the second and third columns the summary of Paris’s law values obtained for different stress ratios (see the first column). Comparing the results obtained for a stress ratio of 0.1 with the tests carried out in [11], the slope value *m* was slightly lower than 2.025 in relation to 2.40. In contrast, slightly higher values of *C* were verified (7.65 $\times \;10^{-8}$ and 4.96 $\times \;10^{-8}$ (mm/cycle) MPa$\sqrt{m}$).

Taking into account that proximity, it was expected that the stress ratio effect be low on the chromium–vanadium spring steel for cracks propagating in regime II. The stress ratio effect on the propagation rate could be explained by the crack closure effect at the crack tip. The average closure ratio, U=0.92, (given by Equation (Equation 10)), indicated that there was only an 8% reduction in the crack tip’s driving force associated with plasticity located at the crack tip, which might explain the low sensitivity to the stress ratio effect on the crack tip. Figure 7 depicts the average values and respective standard deviations measured for the crack closure ratio for stress ratios 0.1 and 0.3.

### 4.3. Critical Stress Intensity Factor

The critical stress intensity factor, $K_{c}$, was calculated by considering the largest crack length measured at failure. The average value found for $K_{c}$ was 138.37±2.61 MPa$\sqrt{m}$ (see Table 5). Considering the condition $B=2.5{\left[K_{Ic}/\sigma_{y}\right]}^{2}$, with a yield strength of 1271.48 MPa and a thickness of 10 mm, the determined value of $K_{c}$ = 138.37 MPa$\sqrt{m}$ did not correspond to the plane strain fracture toughness ($K_{Ic}=80.42$ MPa$\sqrt{m}$). Moreover, in the literature, usual $K_{Ic}$ values for high-strength steels can reach 65 MPa$\sqrt{m}$ [38,39] and steels with a Rockwell C hardness of 45 HRC $K_{Ic}$ can assume values between 50 and 70 MPa$\sqrt{m}$ [13].

### 4.4. Threshold Stress Intensity Factor Range

The procedure for the determination of $\Delta K_{th}$ is illustrated in Figure 8. According to the gathered data presented in Figure 8, $\Delta K_{th}$ is the value of ΔK when $N/N_{th}=1$, which corresponds to the crack length that stops to propagate; this resulted in a $\Delta K_{th}$ of 6.92 MPa$\sqrt{m}$ for $R_{\sigma }=0.1$ (very close to the value observed in [40] of 5 MPa$\sqrt{m}$), 5.78, and 5.39 MPa$\sqrt{m}$ for $R_{\sigma }=0.1,0.3,$ and 0.5, respectively. In [41,42], the same steel grade was tested for $R_{\sigma }=0.1$ (with a 10 mm thick specimen), for a martensitic material (without tempering) and for a steel grade with a ferrite/pearlitic microstructure. The obtained values of 4 and 11.2 MPa$\sqrt{m}$ in [41,42] for martensitic and ferritic/pearlitic microstructure, respectively, validate the obtained result (6.92 MPa$\sqrt{m}$) for our material.

Once the threshold value was determined, it was possible to estimate a power-law curve to predict the da/dN−ΔK relationship in regime I, as written in Equation (Equation 3). The regression method applied in Equation (Equation 3) resulted in $A_{th}\left(R_{\sigma }=0.1\right)$ = 1.5225 $\times \;10^{-6}$, $A_{th}\left(R_{\sigma }=0.3\right)$ = 1.6122 $\times \;10^{-6}$, and $A_{th}\left(R_{\sigma }=0.5\right)$ = 2.1803 $\times \;10^{-6}$ (mm/cycle) MPa$\sqrt{m}$, for the three stress ratios, respectively. Regarding the exponents’ regressors, $p_{th}\left(R_{\sigma }=0.1\right)$ = 1.0112, $p_{th}\left(R_{\sigma }=0.3\right)$ = 0.9321, and $p_{th}\left(R_{\sigma }=0.5\right)$ = 0.7310 (summarized in Table 5). The average values for the power-law curve were $\Delta K_{th}=6.0308\pm 0.7933$ MPa$\sqrt{m}$, $A_{th}$ = 1.7717 $\times \;10^{-6}$± 3.5671 $\times \;10^{-7}$ (mm/cycle) MPa$\sqrt{m}$ and $p_{th}=0.8914\pm 0.1444$, respectively.

From the comparison between $A_{th}$ and $p_{th}$ obtained for different stress ratios $R_{\sigma }$ in regime I, as illustrated in Figure 9, for lower $R_{\sigma }$ values, the slope of the crack propagation rate tended to be greater. Nevertheless, for higher $R_{\sigma }$ ratios, in addition to the crack starting propagating for lower $\Delta K_{th}$ values, after the propagation began, the crack always had a superior crack propagation rate. Graphically, Figure 9 shows that there was a good fitting in the initial zone after the crack began propagating. However, for $\Delta K_{th}$ greater than 10 MPa$\sqrt{m}$, there was an increase in the crack propagation rate. This deviation might be associated with the beginning of the propagation regime II, whereby Equation (Equation 3) is no longer valid.

### 4.5. Global Fatigue Crack Propagation Model

The results previously presented showed a low crack closure effect for the different analysed stress ratios. Considering these results, a global fatigue crack propagation model considering the different stress ratios was then calibrated. Initially, the data were calibrated according to Paris’s law (Equation (Equation 1)), resulting in the curve illustrated in Figure 6. The calibration of the parameters for Equation (Equation 1) resulted in a coefficient *C* = 5.99$\;\times \;10^{-8}$ and an exponent m=2.10, with a coefficient of determination $R^{2}$ of 0.98. Visually, it is verified in Figure 10 that the points of greatest scatter are associated with the zone with the highest crack growth rate for most stress intensity ratios. Considering this scatter, Figure 10 also presents the respective straight-line confidence bands for one standard deviation of da/dN.

Although the model presents an average curve for the stress ratios analysed, the Walker model given by Equation (Equation 4) is normally used to represent the fatigue crack growth as a function of the stress ratio. Usually, the γ parameter is determined using data in propagation regime II; however, as the specimen tested under $R_{\sigma }=0.5$ exhibited a lower crack propagation rate when compared to $R_{\sigma }=$ 0.3 and 0.1, the Walker parameter, γ, was impossible to obtain from (Equation 4). Thus, Equation (Equation 8) was considered to determine γ. The described linear least-squares regression method was considered ($x_{i}=\log\left(1-R_{\sigma ,i}\right)$ and $y_{i}=\log\Delta K_{th,i}$), resulting in γ=0.5767, $\Delta K_{th(R=0.0)}=7.0578$ MPa$\sqrt{m}$ with $R^{2}=0.9013$. Extrapolating Equation (Equation 8) for stress ratio values of −1, we obtained a value of $\Delta K_{th(R=-1)}=9.464$ MPa$\sqrt{m}$, which was very close to the result obtained in [10] of $\Delta K_{th(R=-1)}=10.769$ MPa$\sqrt{m}$, validating the accuracy of the model in Equation (Equation 8) and the results obtained.

Figure 11 illustrates the regression obtained from Equation (Equation 8) and the respective propagation threshold points previously obtained. Comparing the experimental γ with the one obtained via Equation (Equation 5), which resulted in γ = 0.5941 for $\sigma_{uts}=1438.35$ MPa, it was verified that the value of 0.5767 determined from Equation (Equation 8) was in satisfactory agreement.

Once γ was determined, the Walker model written in Equation (Equation 4) resulted via the least-squares method in a coefficient $C_{w,II}$ = 3.47 $\times \;10^{-8}$ and an exponent $m_{w,II}$ = 2.16, with a coefficient of determination $R^{2}=0.89$, as shown in Figure 12. Figure 12 shows a greater scatter in the data obtained for ratios of 0.5 and 0.7; however, all points were contained in the grey area (illustrated in darker blue in Figure 12), corresponding to the prediction curves for one standard deviation of the propagation rate, $\sigma_{da/dN}$, with regards to the average value, $\mu_{da/dN}$. The red and blue areas correspond to predictions considering once and twice the standard deviation of parameters *C* and *m*, respectively denoted as $\sigma_{C}$ and $\sigma_{m}$. Data are presented in Table 6.

Extending Walker’s model (Equation 4) up to regime I (see Equation (Equation 7)) resulted in $C_{w,I,II}$ = 5.78 $\times \;10^{-8}$ and $m_{w,I,II}=1.43$ with a coefficient of determination $R^{2}=0.91$, as illustrated in Figure 13. It turned out that the stress intensity ratio data of 0.1, 0.3, and 0.5 for higher strain intensity ranges were better fitted by Equation (Equation 9), except for a ratio of 0.7. However, as the stress intensity factor range decreased, there was a greater deviation in the experimental results concerning the mode in Equation (Equation 9). Also, Figure 13 suggests that the conservative prediction curve should be given by the mean curve plus one standard deviation for both lower and higher strain intensity ranges instead of the prediction curves considering once and twice the standard deviation of $C_{w,I,II}$ and $m_{w,I,II}$.

The data referring to the models in Equations (Equation 4) and (Equation 9) are presented in Table 6 together with the results obtained by a regression on the global model given by Paris’s law (Equation (Equation 1)). According to the data presented in Table 6, it can be seen that the value of *m* given by Walker’s model considering only the propagation region II is closer to the value obtained by Paris’s law with the global data. However, for the *C* parameter, Walker’s model considering the zone close to the propagation threshold is closer than that obtained by the Paris law.

### 4.6. Fracture Surface Analysis

SEM technology allowed us to analyse the fracture surfaces from the initiation up to the unstable fracture zones. Four samples corresponding to a stress ratio of 0.1 and 0.5 according to the LT and TL propagation systems were considered for the analysis. Figure 14 illustrates an initial pre-crack of around 1 mm in front of $a_{o}$. According to Figure 14, no significant changes in the crack surface’s transition between pre-crack and the fatigue crack growth testing are visible, which is an excellent indication of the proper selection of the pre-cracking loading level.

The effect of the propagation direction of the CT specimens was analysed by considering two samples (manufactured in the LT and TL directions) tested at an $R_{\sigma }$ of 0.1 (see Figure 15). In general, there were no significant differences to point out between the specimens. Both specimens featured some large cleavage facets with identical dimensions. Furthermore, micro-cracks could be observed in multiple directions in relation to the propagation direction for both specimens. This similarity in behaviour was then reflected in the material behaviour response in the Paris curves (Figure 6) and in the respective Paris law exponent coefficients (Equation (Equation 1)).

With regard to the stress ratio effect, two specimens tested under a stress ratio of 0.1 and 0.5 were selected. Figure 16 shows the comparison of the crack propagation paths from the pre-crack initiation zone to the initiation moment of the crack’s unstable propagation for a ×100 magnification.

Comparing both fracture surfaces along the crack propagation path, it was verified that the pre-crack zone tended to present a rougher fracture surface, compared to the stable propagation zone, as illustrated in Figure 17 at 1 and 6 mm. The rougher surface at the 1 mm crack length was associated with the test speed, which was higher than the proper testing speed due to the need for a rapid generation of a pre-crack.

Additionally, the stable crack propagation zone from 3 up to 20 mm exhibited a smoother fracture surface. For a lower stress ratio ($R_{\sigma }$ = 0.1), the surface appeared to be slightly rougher in contrast to CT specimens tested at $R_{\sigma }$ = 0.5. After 20 mm, the fracture surface of the specimen tested at $R_{\sigma }$ = 0.1 started to be rougher. This trend is easily visible on the micrographs of Figure 17 at 6 mm. From a crack length of 20 mm in Figure 16, the image begins to indicate a change in the fracture surface topography, becoming rougher in both specimens. This change is associated with the increase in crack propagation rate in regime III, not showing significant differences for different stress ratios as verified in Figure 16 at 21 mm. In this zone (no. 5), the appearance of ductile dimples with small dimensions is quite visible [16].

Increasing the magnification to ×2.00k, one can state that the propagation occurred essentially at the transgranular level, with cleavage micro-mechanisms being the cause of the pre-crack initiation zone, crack propagation, stable, and unstable cracks. For the entire propagation zone analysed, no fatigue striations with dimensions of the order of 10 μm were identified. Figure 18, picture zone 3, shows the existence of a longitudinal crack which was formed by the decohesion of the metallic matrix with the non-metallic slender inclusion. These small crack geometries are often found on specimens manufactured with the TL propagation system.

Additionally, from Figure 16, a greater number of darker spots appeared in the specimen tested at $R_{\sigma }$ = 0.1. These are cleavage facets, indicating that for lower stress ratios, there was a greater likelihood for cleavage facets to form (see Figure 18). Still, for the specimen tested at $R_{\sigma }$ = 0.1, in Figure 18, at 16 mm, we observed the existence of cleavage micro-cracks of approximately 20 μm within the facets. Moving to the unstable crack propagation zone (21 mm), the appearance of ductile dimples was verified for the specimen tested for $R_{\sigma }$ = 0.5.

## 5. Conclusions

The propagation behaviour of fatigue cracks in chromium–vanadium steel was analysed for several stress ratios using CT specimens obtained from the LT and TL directions of suspension spring leaves.The propagation behaviour in regime II was initially analysed according to the Paris law, which verified a low effect of the stress ratio. This low difference could be explained essentially by the high value of the crack closure ratio, U=0.92, and statistically by the coefficient of determination obtained for the global Paris regression model taking into account all stress ratios, $R^{2}=0.98$, for *C* = 5.99 $\times \;10^{-8}$± 1.98 $\times \;10^{-8}$ (mm/cycle) MPa$\sqrt{m}$ and m = 2.10 ± 0.09. The results were monitored until the moment of failure, making it possible to estimate the value of the critical stress intensity factor, resulting in $K_{c}$ = 138.37 ± 2.61 MPa$\sqrt{m}$.

In a second phase, the propagation threshold value was determined for the stress ratios of 0.1, 0.3, and 0.5, verifying a clear effect of the stress ratio associated with the propagation threshold with values of 6.92, 5.78, and 5.39 MPa$\sqrt{m}$, respectively. The propagation law parameters for regime I were determined for each stress ratio, obtaining an average value of $A_{th}$ = 1.77 $\times \;10^{-6}$± 3.57 $\times \;10^{-7}$ MPa$\sqrt{m}$ and $p_{th}$ = 0.89 ± 0.14.

Since the material exhibits a strong influence on the stress intensity factor in propagation regime I, Walker’s model was used to describe crack growth in regimes I and II. The Walker parameter was determined by its relation to the propagation threshold, resulting in γ = 0.5767. Initially, the parameters of the Walker’s model for regime II were considered, such that the value obtained for $C_{w,II}$ and $m_{w,II}$ were, respectively, 3.47 $\times \;10^{-8}$± 1.32 $\times \;10^{-8}$ mm/cycle $\sqrt{m}$ and 2.16 ± 0.09, with $R^{2}=0.89$. Then, Walker’s model contemplating the two propagation regimes was used, resulting in $C_{w,II}$ = 5.78 $\times \;10^{-8}$± 1.43 $\times \;10^{-8}$ (mm/cycle) MPa$\sqrt{m}$ and $m_{w,II}$ = 1.43 ± 0.05, with $R^{2}=0.91$. In fact, the introduction of propagation region I changed the value of *m* from 2.16 to 1.43, moving away from the value obtained for the Paris law and increasing the coefficient *C* from 3.47 $\times \;10^{-8}$ to 5.78 $\times \;10^{-8}$, approaching the value obtained from Paris’s law.

Finally, an analysis of the fracture surfaces of the specimens was carried out. The effect of the manufacturing direction of the specimens and the stress intensity ratio were evaluated. In general, the fracture surface analysis showed that the crack propagation process occurred predominantly in a transgranular way with micro-cleavage processes and without visible fatigue striations. From the analysis conducted on the LT and TL propagation systems, no significant differences were revealed, which was also demonstrated by the coefficients and exponents obtained by the Paris law. However, the presence of slender small cracks, caused by the decohesion of the metallic matrix interface with the non-metallic inclusion, was verified for the TL specimens. Comparing the evaluated stress ratios, it was verified that the fracture surface topographies of specimens tested at $R_{\sigma }$ = 0.1 tended to be rougher than specimens tested at $R_{\sigma }$ = 0.5. In addition, for $R_{\sigma }$ = 0.1, specimens presented a greater number of larger-size cleavage facets, making it possible to observe for some a very smooth surface, which indicated the crack closure occurrence. Other aspects such as micro-cleavage cracks (with 20 μm) and ductile dimples were sporadically observed.

## Figures and Tables

**Figure 1 materials-17-01831-f001:**
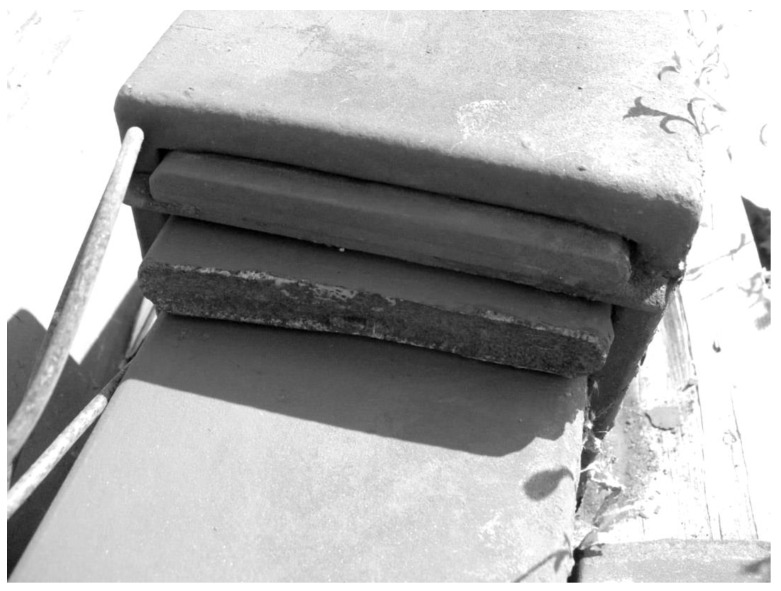
Fatigue fracture of the master spring leaf of a parabolic leaf spring.

**Figure 2 materials-17-01831-f002:**
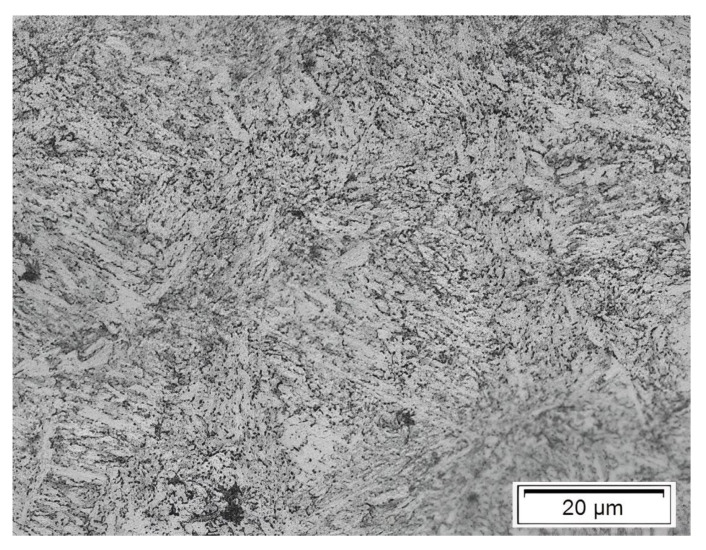
Typical microstructure of the chromium–vanadium alloyed steel for all tested specimens [4].

**Figure 3 materials-17-01831-f003:**
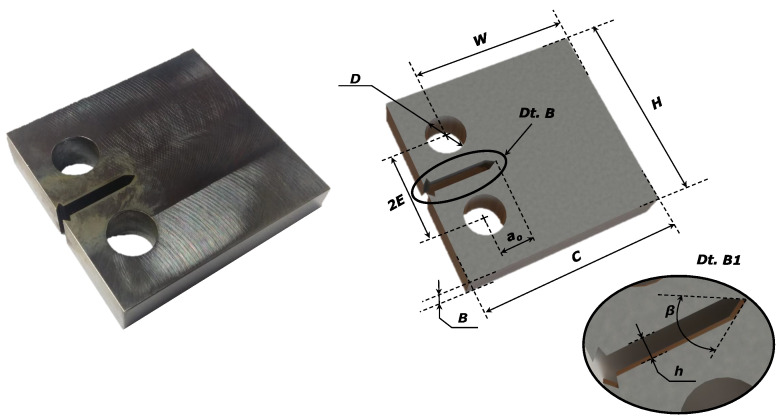
Geometry of the compact tension specimen used to evaluate the crack growth propagation under mode I fatigue loading.

**Figure 4 materials-17-01831-f004:**
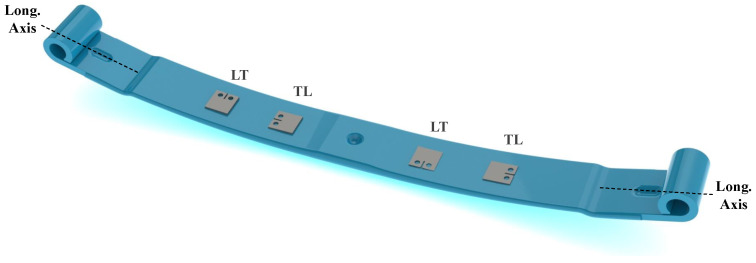
Illustration of the directions from which samples were taken according to the LT and TL labels.

**Figure 5 materials-17-01831-f005:**
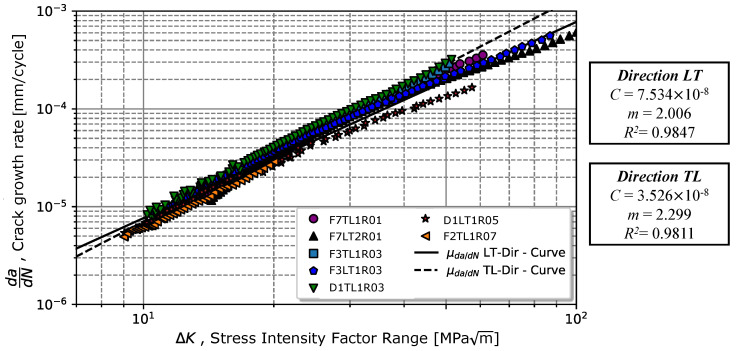
Influence of the rolling direction in the crack propagation rate in propagation regime II.

**Figure 6 materials-17-01831-f006:**
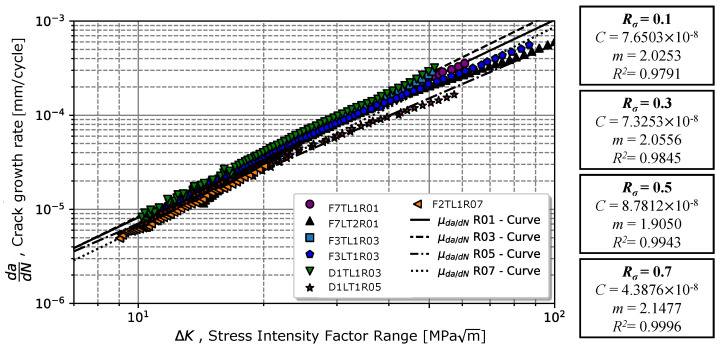
Variation in the crack propagation rate in propagation regime II in relation to the applied stress intensity factor range for different stress intensity ratios.

**Figure 7 materials-17-01831-f007:**
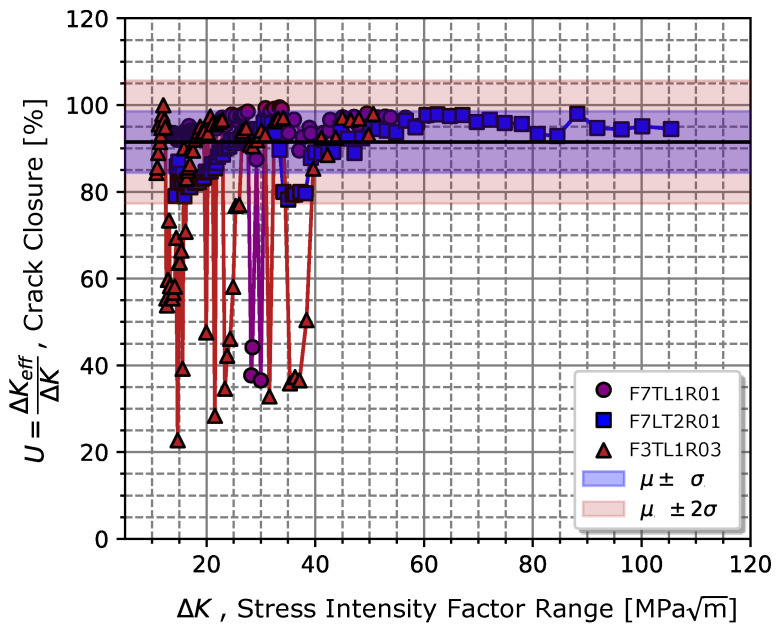
Average value of the crack closure ratio and respective standard deviation determined throughout the fatigue crack propagation tests.

**Figure 8 materials-17-01831-f008:**
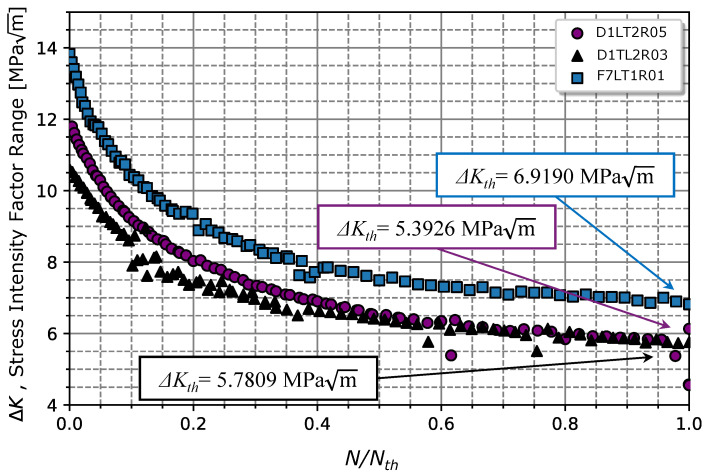
Determination of the threshold of the stress intensity factor range from the analysis of the variation in stress intensity factor range throughout the crack growth fatigue testing.

**Figure 9 materials-17-01831-f009:**
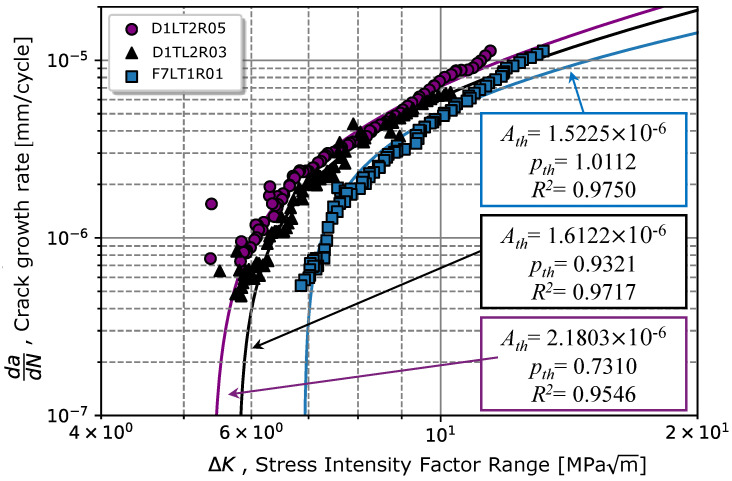
Effect of the stress ratio on the propagation threshold and the crack propagation model for regime I according to Equation (Equation 3).

**Figure 10 materials-17-01831-f010:**
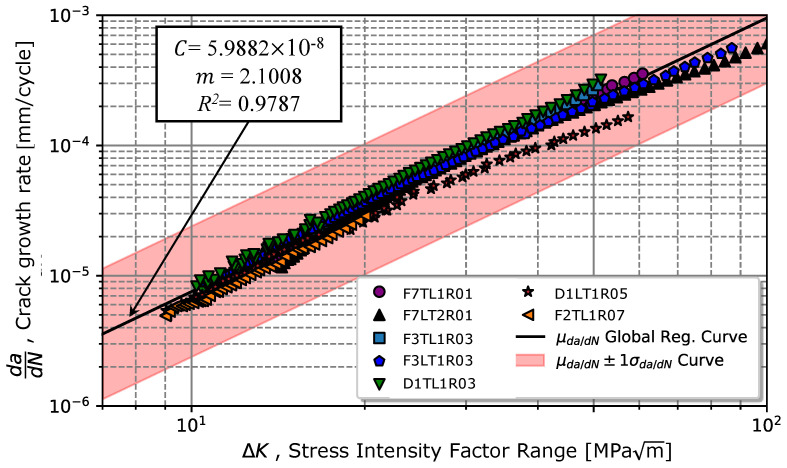
Global da/dN−ΔK curve representing the variation in crack propagation rate in propagation regime II for several stress ratios and its respective standard deviation.

**Figure 11 materials-17-01831-f011:**
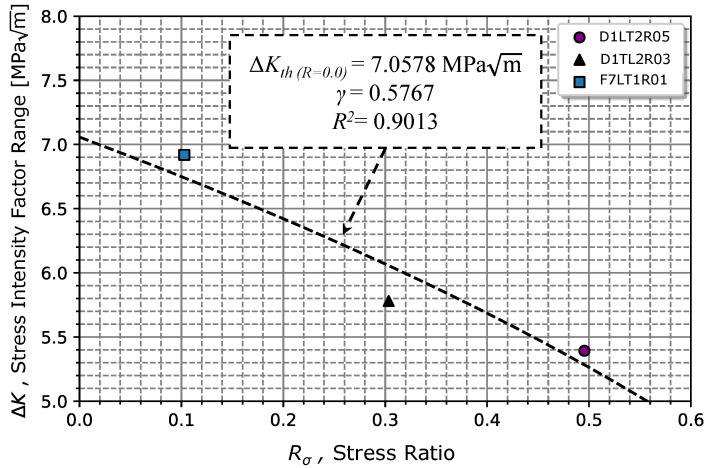
Relationship for determining the Walker’s parameter, γ, in accordance with Equation (Equation 8).

**Figure 12 materials-17-01831-f012:**
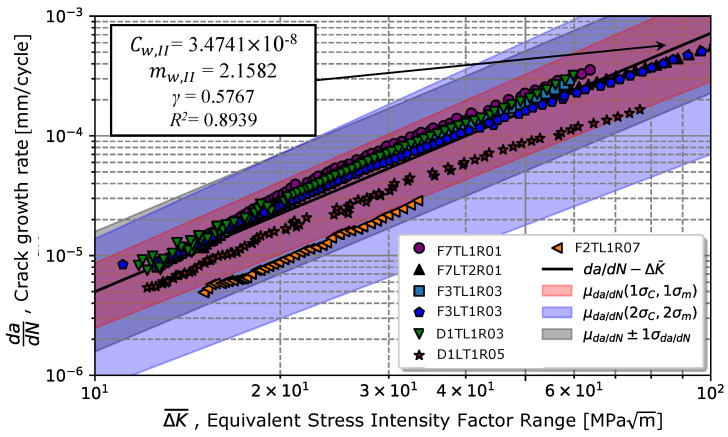
Representation of the Walker’s model (Equation (Equation 4)) for different stress intensity factor ratios.

**Figure 13 materials-17-01831-f013:**
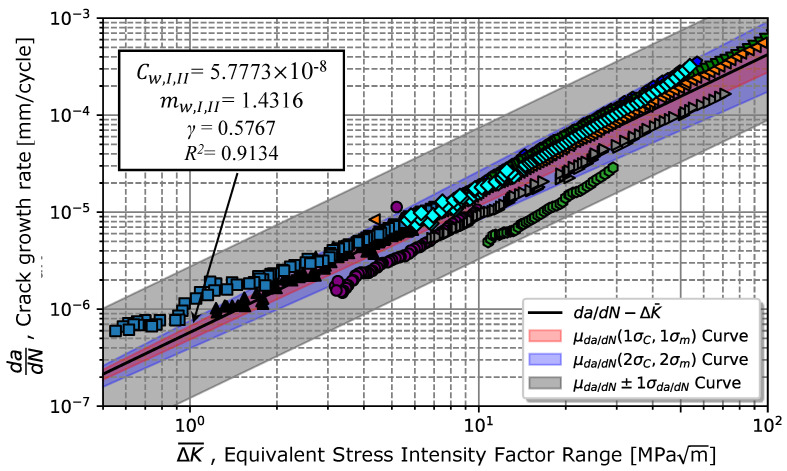
Representation of the Walker’s model (Equation (Equation 9)) for different stress intensity factor ratios.

**Figure 14 materials-17-01831-f014:**
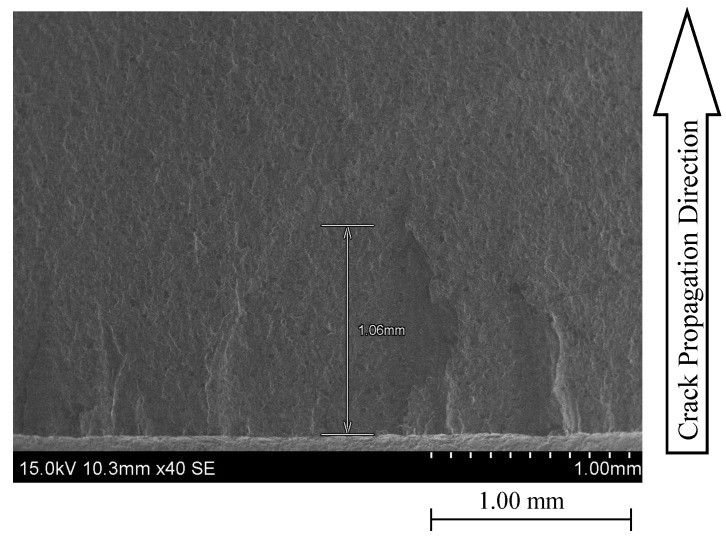
Illustration of a pre-crack size for a ×40 magnification.

**Figure 15 materials-17-01831-f015:**
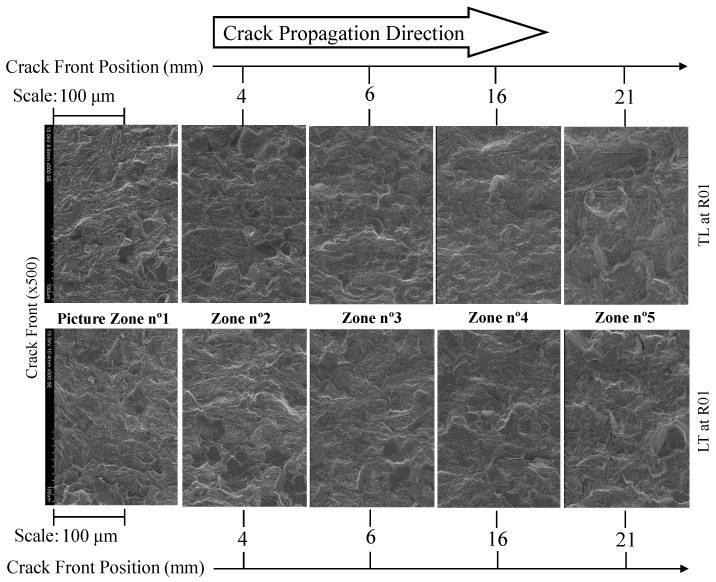
Comparison between fracture surface paths of specimens obtained from LT and TL directions for a ×500 magnification ($R_{\sigma }$ = 0.1).

**Figure 16 materials-17-01831-f016:**
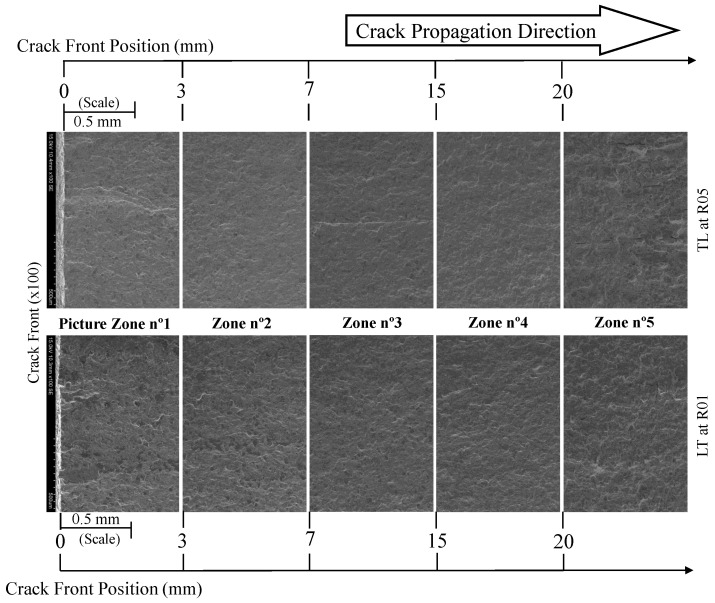
Crack propagation path from the pre-crack initiation zone to the unstable propagation moment for a ×100 magnification ($R_{\sigma }$ = 0.1 and 0.5).

**Figure 17 materials-17-01831-f017:**
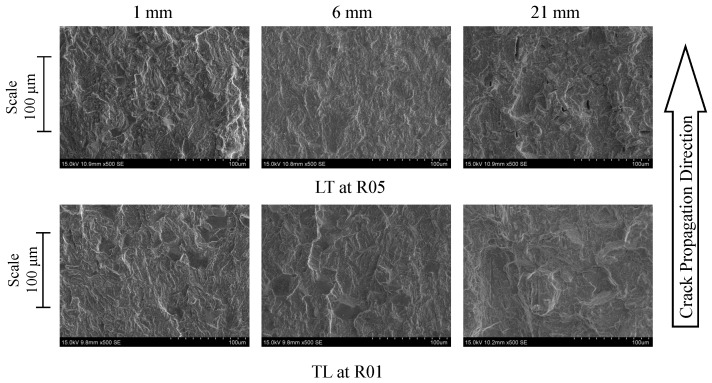
Topography comparison of fracture surfaces from the initiation zone to the unstable propagation zone for $R_{\sigma }$ = 0.1 and 0.5.

**Figure 18 materials-17-01831-f018:**
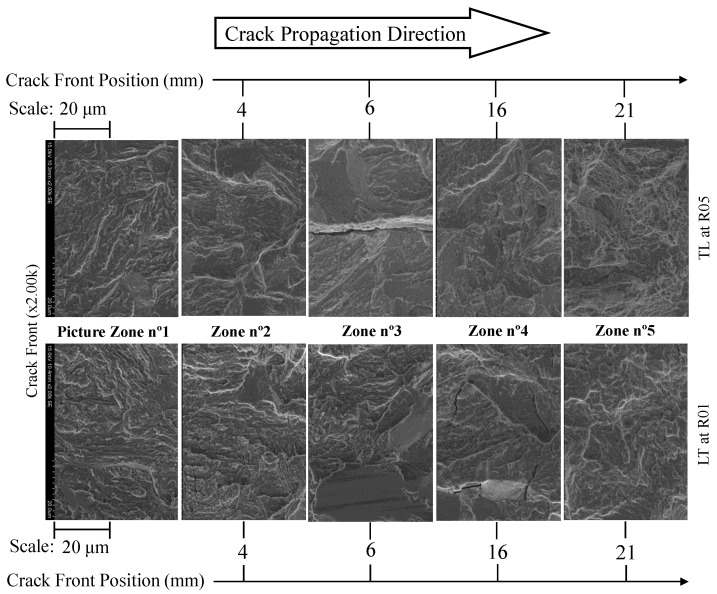
Magnification of the crack propagation path from the pre-crack initiation zone to the beginning moment of unstable propagation for a ×2.00k magnification ($R_{\sigma }$ = 0.1 and 0.5).

**Table 1 materials-17-01831-t001:** Standard chemical composition of 51CrV4 steel grade in % wt.

Material	C	Si	Mn	Cr	V	S	Pb	Fe
51CrV4 (1.815)	0.47–0.55	≤0.40	0.70–1.10	0.90–1.20	≤0.10–0.25	≤0.025	≤0.025	96.45–97.38

**Table 2 materials-17-01831-t002:** Monotonic mechanical properties obtained from the chromium–vanadium alloyed steel,
51CrV4 [4].

	*E*	$\sigma_{y}$	$\sigma_{\mathrm{uts}}$	$\varepsilon_{f}$	$R_{A_{f}}$
	[GPa]	[MPa]	[MPa]	[%]	[%]
Average	200.54	1271.48	1438.35	7.53	34.69
Std. Dev. []	±6.02	±53.32	±73.84	±0.77	±10.39
DIN 51CrV4 (1.8159)	200	1200	1350–1650	6	30

**Table 3 materials-17-01831-t003:** Average dimensions of CT specimens used in fatigue crack growth testing according to the ASTM E647 standard [36].

$a_{o}$ [mm]	*W* [mm]	*B* [mm]	*H* [mm]	*C* [mm]	*h* [mm]	*D* [mm]	*d* [mm]	α [deg]
10.20	35.04	9.95	47.99	49.98	2.56	21.95	10.01	60
± 0.31	± 0.07	± 0.04	± 0.03	± 0.03	± 0.09	± 0.10	± 0.04	

**Table 4 materials-17-01831-t004:** Comparison of Paris’s law parameters for LT and TL propagation directions.

$R_{\sigma }$	*C* (LT) [(mm/cycle) MPa$\sqrt{m}$]	*C* (TL) [(mm/cycle) MPa$\sqrt{m}$]	*m* (LT)	*m* (TL)
0.1	8.8364 $\times \;10^{-8}$	4.1781 $\times \;10^{-8}$	1.9653	2.2343
0.3	8.3013 $\times \;10^{-8}$	4.2025 $\times \;10^{-8}$	2.0087	2.2522
0.5	8.7819 $\times \;10^{-8}$	-	1.9050	-
0.7	5.9891 $\times \;10^{-8}$	-	1.7109	-
Average	7.534 $\times \;10^{-8}$	3.526 $\times \;10^{-8}$	2.006	2.299
± Std.	3.761 $\times \;10^{-8}$	1.827 $\times \;10^{-8}$	0.1249	0.1377

**Table 5 materials-17-01831-t005:** Property values that quantify the propagation phase in regimes I, II, and III for different stress ratios.

	*C*		$K_{\mathrm{Ic}}$	$a_{c}$	$\Delta K_{\mathrm{th}}$	$A_{\mathrm{th}}$	
$R_{\sigma }$	[(mm/cycle) MPa$\sqrt{m}$]	m	[MPa$\sqrt{m}$]	[mm]	[MPa$\sqrt{m}$]	[(mm/cycle) MPa$\sqrt{m}$]	$p_{\mathrm{th}}$
0.1	7.6503 $\times \;10^{-8}$	2.0253	137.57	28.94	6.919	2.180 $\times \;10^{-6}$	0.7310
0.3	7.3253 $\times \;10^{-8}$	2.0556	139.86	29.00	5.781	1.612 $\times \;10^{-6}$	0.9310
0.5	8.781 $\times \;10^{-8}$	1.9050	134.48	29.03	5.393	1.523 $\times \;10^{-6}$	1.0112
0.7	4.3876 $\times \;10^{-8}$	2.1477	137.97	28.82	-	-	-
Average	5.9882 $\times \;10^{-8}$	2.1008	138.37	28.95	6.0308	1.7717 $\times \;10^{-6}$	0.891
± Std.	1.9760 $\times \;10^{-8}$	0.0910	2.61	0.08	0.7933	3.5671 $\times \;10^{-7}$	0.1444

**Table 6 materials-17-01831-t006:** Comparison of the properties that identify the propagation phase in regimes I and II given by the models in Equations (Equation 1), (Equation 4) and (Equation 9).

γ	Equation		*C* [(mm/cycle) MPa$\sqrt{m}$]	*m*	$R^{2}$
0.5767	Walker (Equation 4)	Average	3.4741 $\times \;10^{-8}$	2.1582	0.8939
		± Std.	1.3215 $\times \;10^{-8}$	0.0978	
	Walker (Equation 9)	Average	5.7773 $\times \;10^{-8}$	1.4316	0.9134
		± Std.	1.4316 $\times \;10^{-8}$	0.0548	
N.D	Paris (Equation 1)	Average	5.9882 $\times \;10^{-8}$	2.1008	0.9787
		± Std.	1.9760 $\times \;10^{-8}$	0.0910	

## Data Availability

The data presented in this study are available on request from the corresponding author. The data are not publicly available due to privacy restrictions.

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
