# Peer review of "Fatigue Crack Propagation of 51CrV4 Steels for Leaf Spring Suspensions of Railway Freight Wagons"

_materials, 2024, doi:10.3390/ma17081831_

Round 1
Reviewer 1 Report
Comments and Suggestions for Authors
1. The authors used a rather strange system for marking text fragments. They have a separate introduction, a separate section 2, in which previously proposed equations for various stages of fatigue crack propagation are analyzed.
2. Section 2 practically does not indicate the validity of the models proposed by various authors for spring steels.
3. The authors widely use the terms “mode I” and “mode II” without indicating what they are.
4. All results were obtained only for steel of one heat and one heat treatment mode. The possibility of changing fatigue characteristics when changing, for example, tempering temperature and transition to tempering troostite structures has not been assessed in any way.
5. The authors do not indicate how the groove was cut in the sample and the degree of its sharpness before the formation of a fatigue crack.
6. The authors do not explain the marking system for samples of type F7TL1R01.
Author Response
- The authors used a rather strange system for marking text fragments. They have a separate introduction, a separate section 2, in which previously proposed equations for various stages of fatigue crack propagation are analyzed.
A: The following separation of the two topics into two sections was chosen to facilitate the reading of the document, in such a way that in section 1, the introduction and summary of the document are described, and in section 2, only the writing relating to the theory and analytical models considered for the analysis.
- Section 2 practically does not indicate the validity of the models proposed by various authors for spring steels.
A: The models presented in this scientific article are general models that are suitable and widely used for predicting fatigue crack propagation in various metals. However, in order to justify its application, the different models applied are justified for spring steels, by citing several studies.
References: 10,12,22,23,24
- The authors widely use the terms “mode I” and “mode II” without indicating what they are.
A: Mode I and Mode II are well-defined in studies regarding fatigue crack propagation studies. However, I add in brackets the meaning of “mode I” – (crack propagation perpendicular to the loading)
- All results were obtained only for steel of one heat and one heat treatment mode. The possibility of changing fatigue characteristics when changing, for example, tempering temperature and transition to tempering troostite structures has not been assessed in any way.
A: The effect of tempering temperature on fatigue resistance, despite affecting its resistance, was not evaluated, because leaf springs must be tempered at a certain temperature (standardized conditions). However, to clarify the reader's analysis of the document, the following sentence was added to the text: “The steel under investigation is the chromium-vanadium alloyed steel 51CrV4 with an average carbon content of roughly 0.50 % as presented in Table 1. Being this steel grade standardised to be quenched at 850 oC (40 min) in an oil bath, and then tempered at 450 oC for 90 min, the 51CrV4 steel (as received) exhibited a tempered martensite microstructure with retained austenite (white phases) as shown in Figure 2.”
- The authors do not indicate how the groove was cut in the sample and the degree of its sharpness before the formation of a fatigue crack.
A: The surfaces of the specimens were milled. This information was added to the paper: “Compact tension (CT) specimens are manufactured according to the guidelines ASTM E647 standard, resulting in the geometry specimen with a milled surface illustrated in Figure 3. ”
- The authors do not explain the marking system for samples of type F7TL1R01.
The following text was added: “The sample marking system is in accordance with the batch and leaf from which the sample (first and second character) was taken, the crack propagation directions, TL and LT, are identified by the third and fourth characters. The fifth parameter is used to identify samples obtained by the same batch, same crack propagation direction and tested under the same stress ratio conditions. The last three characters (sixth to octave) are identifiers of the stress ratio used in the test (stress ratio of 0.1, 0.3, 0.5 and 0.7).”
Reviewer 2 Report
Comments and Suggestions for Authors
Manuscript describes fatigue crack propagation and some characterization methods and prediction models to assist design and maintenance engineers in understanding fatigue behaviour in the initiation and propagation phase of cracks in leaf springs for railway freight wagons to improve quality of these materials
- In the abstract and the introduction mentioned “51CrV4 spring” try to describe it, because not all readers know this compound
- Introduction is too short (lines 17-33), lines 34 to 54 described experimental part, improve the introduction according international publication
- 3.1. Chemical Composition and Microstructure, include information where purchase, Aldrich or somewhere, city, country, etc. Including sentence “all materials were used as received” or some similar sentence
- Why is important to include Figure 5, maybe can move to supplementary materials
- Manuscript has 20 Figures and 6 Tables, Tables maybe is adequate but 20 Figures are many, maybe 10 Figures will be Ok. Try to move some Figures and/or Tables to supplementary materials
- Discussion part needs to improve
Author Response
Manuscript describes fatigue crack propagation and some characterization methods and prediction models to assist design and maintenance engineers in understanding fatigue behaviour in the initiation and propagation phase of cracks in leaf springs for railway freight wagons to improve quality of these materials
- In the abstract and the introduction mentioned “51CrV4 spring” try to describe it, because not all readers know this compound
A: Replaced in the abstract 51CrV4 spring steel by chromium-vanadium alloyed steel.
“The aim of this document is precisely to characterise the mechanical crack growth behaviour of the chromium-vanadium alloyed steel representative of leaf springs under cyclic conditions, that is, crack propagation in mode I.”
- The introduction is too short (lines 17-33), lines 34 to 54 described experimental part, improve the introduction according international publication.
A: The introduction was improved.
- Chemical Composition and Microstructure, include information where purchase, Aldrich or somewhere, city, country, etc. Including sentence “all materials were used as received” or some similar sentence
A: The following paragraph was changed: “The steel under investigation is the chromium-vanadium alloyed steel 51CrV4 with an average carbon content of roughly 0.50 % as presented in Table 1. Being this steel grade standardised to be quenched at 850 oC (40 min) in an oil bath, and then tempered at 450 oC for 90 min, the 51CrV4 steel (as received) exhibited a tempered martensite microstructure with retained austenite (white phases) as shown in Figure 2.”
And caption in Figure 2. “Typical microstructure of the chromium-vanadium alloyed steel for all tested specimen.”
- Why is important to include Figure 5, maybe can move to supplementary materials
A: Figure 5 was removed, and the following text was added: “The fatigue crack propagation tests were carried out in a MTS 810 testing machine equipped with a MTS clevis gripping system to measure the crack opening displacement.”
- Manuscript has 20 Figures and 6 Tables, Tables maybe is adequate but 20 Figures are many, maybe 10 Figures will be Ok. Try to move some Figures and/or Tables to supplementary materials.
A: Two Figures was removed (Figures 4 and 8)
- Discussion part needs to improve
A: Discusion was improved comparing obtained data with existing data in the literature.
Reviewer 3 Report
Comments and Suggestions for Authors
The primary question addressed in the research revolves around understanding the fatigue crack propagation behavior of 51CrV4 spring steel, which represents leaf springs commonly utilized in railway freight wagons. Specifically, the study delves into how cracks propagate under cyclic loading conditions (mode I) and aims to establish fatigue crack growth prediction models, such as the Paris and Walker models. These models take into account various factors including stress ratio, propagation threshold, critical stress intensity factor, and crack closure ratio.
The research is notable for its originality and relevance in the field. It fills a specific gap by focusing on the fatigue behavior of a crucial component in railway safety: leaf springs in freight wagons. By offering insights into crack propagation behavior and developing predictive models, the study provides practical assistance to design and maintenance engineers, thereby enhancing the safety and reliability of railway components.
Moreover, the research conducts a comprehensive analysis of fracture surfaces, which contributes to a deeper understanding of the crack propagation process. Overall, the findings of this research are valuable to the field of railway engineering and materials science, offering practical implications for the design, maintenance, and safety of railway transportation systems.
While the references are relevant, it is suggested that the authors consider expanding the list to include a more comprehensive review of existing literature. This can strengthen the theoretical foundation of the research and demonstrate a thorough understanding of the field. Incorporating additional references from related studies, advancements, or alternative methodologies would contribute to the robustness of the paper. It is essential to acknowledge and build upon the existing body of knowledge in the field to situate the current research within the broader scientific context.
I have detected an issue on the SEM images, which I think is an important problem to be addressed.
I do not understand the “100 mm” scale bar under the microscopic image of Figure 16. The internal scale bar shows 1 mm for the same length. The vertical arrow is also about 1 mm long. Thus, I would recommend either to completely remove the “100 mm” scale bar, or if I misunderstood something, please explain why this 100-times difference is shown. It is not clear from the figure caption and from the main text either.
On Figure 18 the SEM embedded scalebar shows 500 um about the same length as the drawn scalebar of 1 mm above and below the left subpictures.
On Figure 20 the caption says “x200k magnification”. This would be 2000-times more magnification that what is referred on Figure 18 (x100). Here the drawn 20 um scalebar is about 1.27-times longer than the embedded 20 um scalebar. In other words, if I assume that the embedded scalebar is correct on Fig 18 (and the drawn 1 mm is wrong, it should be 500 um), than the embedded scalebar on Fig 20 calculates 20-times more (= x2000 = x2k) magnification than that of fig 18.
Therefore, I recommend recalculating ALL scalebars, or remove all of them, an use only the SEM embedded ones. If this makes it necessary, the conclusions should be reconsidered.
Due to these issues, I recommend a major revision.
There are also some minor practical notes:
In line 125 the citation is missing: [??]
In line 222 one of the scientific notation should be edited:
C = 7.534E − 08 ± 3.761E-08
I.e. the first E should not be Italic, and the minus sign should be a short one. Spaced should also be removed. This way “7.534E – 08” will become similar to “3.761E-08”, and other scientific notations, e.g. lines 220-221, and in Table 4.
The same applies to “7.65E − 08” in line 236, “7.33E − 08” in line 237, “4.39E − 08” in line 239, “1.5225E − 06” in line 265, “1.6122E – 06” and “2.1803E − 06” in line 266, “1.7717E – 07” and “3.5671E − 07” in line 269, “5.99E − 08” in line 285.
Author Response
The primary question addressed in the research revolves around understanding the fatigue crack propagation behavior of 51CrV4 spring steel, which represents leaf springs commonly utilized in railway freight wagons. Specifically, the study delves into how cracks propagate under cyclic loading conditions (mode I) and aims to establish fatigue crack growth prediction models, such as the Paris and Walker models. These models take into account various factors including stress ratio, propagation threshold, critical stress intensity factor, and crack closure ratio.
The research is notable for its originality and relevance in the field. It fills a specific gap by focusing on the fatigue behavior of a crucial component in railway safety: leaf springs in freight wagons. By offering insights into crack propagation behavior and developing predictive models, the study provides practical assistance to design and maintenance engineers, thereby enhancing the safety and reliability of railway components.
Moreover, the research conducts a comprehensive analysis of fracture surfaces, which contributes to a deeper understanding of the crack propagation process. Overall, the findings of this research are valuable to the field of railway engineering and materials science, offering practical implications for the design, maintenance, and safety of railway transportation systems.
- While the references are relevant, it is suggested that the authors consider expanding the list to include a more comprehensive review of existing literature. This can strengthen the theoretical foundation of the research and demonstrate a thorough understanding of the field. Incorporating additional references from related studies, advancements, or alternative methodologies would contribute to the robustness of the paper. It is essential to acknowledge and build upon the existing body of knowledge in the field to situate the current research within the broader scientific context.
A: Literature review (introduction section) was improved.
- I have detected an issue with the SEM images, which I think is an important problem to be addressed.
- I do not understand the “100 mm” scale bar under the microscopic image of Figure 16. The internal scale bar shows 1 mm for the same length. The vertical arrow is also about 1 mm long. Thus, I would recommend either to completely remove the “100 mm” scale bar, or if I misunderstood something, please explain why this 100-times difference is shown. It is not clear from the figure caption and from the main text either.
A: The actual scale of each of the figures is identified as 0.5mm, 100um and 20um. The scale in “mm” only refers to the crack front position to which each of the picture zones. For a better understanding of the figures, these figures were edited and corrected according to the comments provided by the reviewer.
- On Figure 18 the SEM embedded scalebar shows 500 um about the same length as the drawn scalebar of 1 mm above and below the left subpictures.
- On Figure 20 the caption says “x200k magnification”. This would be 2000-times more magnification that what is referred on Figure 18 (x100). Here the drawn 20 um scalebar is about 1.27-times longer than the embedded 20 um scalebar. In other words, if I assume that the embedded scalebar is correct on Fig 18 (and the drawn 1 mm is wrong, it should be 500 um), than the embedded scalebar on Fig 20 calculates 20-times more (= x2000 = x2k) magnification than that of fig 18.
Therefore, I recommend recalculating ALL scalebars, or remove all of them, an use only the SEM embedded ones. If this makes it necessary, the conclusions should be reconsidered.
Due to these issues, I recommend a major revision.
There are also some minor practical notes:
- In line 125 the citation is missing: [??]
A: The citations were introduced.
- In line 222 one of the scientific notation should be edited:
C = 7.534E − 08 ± 3.761E-08
I.e. the first E should not be Italic, and the minus sign should be a short one. Spaced should also be removed. This way “7.534E – 08” will become similar to “3.761E-08”, and other scientific notations, e.g. lines 220-221, and in Table 4.
The same applies to “7.65E − 08” in line 236, “7.33E − 08” in line 237, “4.39E − 08” in line 239, “1.5225E − 06” in line 265, “1.6122E – 06” and “2.1803E − 06” in line 266, “1.7717E – 07” and “3.5671E − 07” in line 269, “5.99E − 08” in line 285.
A: All syntax errors were modified to the format “C = 7.534E−08 ± 3.761E-08”.
Round 2
Reviewer 3 Report
Comments and Suggestions for Authors
I looked through the paper again, and concluded that it was nicely corrected. Thus I recommend accepting it in the present form.